# The Role of PK/PD Analysis in the Development and Evaluation of Antimicrobials

**DOI:** 10.3390/pharmaceutics13060833

**Published:** 2021-06-03

**Authors:** Alicia Rodríguez-Gascón, María Ángeles Solinís, Arantxa Isla

**Affiliations:** 1Pharmacokinetic, Nanotechnology and Gene Therapy Group (PharmaNanoGene), Centro de investigación Lascaray ikergunea, Faculty of Pharmacy, University of the Basque Country UPV/EHU, Paseo de la Universidad 7, 01006 Vitoria-Gasteiz, Spain; alicia.rodriguez@ehu.eus (A.R.-G.); marian.solinis@ehu.eus (M.Á.S.); 2Bioaraba, Pharmacokinetic, Nanotechnology and Gene Therapy Group (PharmaNanoGene), 01006 Vitoria-Gasteiz, Spain

**Keywords:** antibiotic, pharmacokinetics, pharmacodynamics, antimicrobial resistance

## Abstract

Pharmacokinetic/pharmacodynamic (PK/PD) analysis has proved to be very useful to establish rational dosage regimens of antimicrobial agents in human and veterinary medicine. Actually, PK/PD studies are included in the European Medicines Agency (EMA) guidelines for the evaluation of medicinal products. The PK/PD approach implies the use of in vitro, ex vivo, and in vivo models, as well as mathematical models to describe the relationship between the kinetics and the dynamic to determine the optimal dosing regimens of antimicrobials, but also to establish susceptibility breakpoints, and prevention of resistance. The final goal is to optimize therapy in order to maximize efficacy and minimize side effects and emergence of resistance. In this review, we revise the PK/PD principles and the models to investigate the relationship between the PK and the PD of antibiotics. Additionally, we highlight the outstanding role of the PK/PD analysis at different levels, from the development and evaluation of new antibiotics to the optimization of the dosage regimens of currently available drugs, both for human and animal use.

## 1. Introduction

Antimicrobials—including antibiotics, antivirals, antifungals, and antiparasitics—are essential medicines needed for a basic health-care system [1]. The World Health Organization (WHO) has declared antimicrobial resistance (AMR) as one of the top 10 global public health threats facing humanity. The AMR phenomenon is a serious and globally extended challenge that occurs when bacteria, viruses, fungi, and parasites change over time and no longer respond to medicines, threatening the effective prevention and treatment of an ever-increasing range of infections [2].

AMR increases the risk of disease spread, severe illness, death, and it has a significant economic impact, compromising not only the ability to treat infectious diseases, but also impeding many other advances in the field of medicine. Misuse and overuse of antimicrobials are the main drivers in AMR development. In this context, WHO released the Access, Watch, and Reserve (AWaRe) classification, which categorizes the antibiotics into different stewardship groups to emphasize the importance of their appropriate use, with the aim of supporting the development of tools for antibiotic management and to reduce bacterial resistance [3]. Moreover, in May 2015, the World Health Assembly endorsed a global action plan to face AMR, including antibiotic resistance, the most urgent drug resistance trend. The optimization of the use of antimicrobial agents is one of the five strategic objectives set out to achieve this goal [4]. This objective is especially relevant considering that in the last 10 years, no new group of antibiotics has been marketed in Europe [5].

AMR is a cross-cutting issue, and therefore, for achieving optimal health and well-being outcomes it has to be tackled from a One Health perspective, recognizing the interconnections between people, animals, plants, and their shared environment [6]. In this sense, the coronavirus disease 2019 (COVID-19) pandemic had increased awareness of the One Health approach to prevent the emergence of infectious diseases, strengthen the need to proceed with this approach at all levels of academia, research, and governments around the world [7]. Nowadays, the need for a One Health approach to address AMR is supported internationally and included in the action plans of many countries [8]. 

Focusing on antibiotherapy, an ideal treatment optimization requires information on the mechanisms involved in the effect of the antibiotics (pharmacodynamics, PD) and the evolution of the antibiotic concentration in the patients (pharmacokinetics, PK). Different models, including in vitro, ex vivo, and in vivo, have been developed to establish the relationship between the PK and the PD of antimicrobials, and one of the main applications is to design dosing strategies to enhance the probability of success of the antibiotic therapy, as well as minimize the side effects and the emergence of resistances [9]. Moreover, reminding that One Health aims to improve the assessment, treatment, and prevention of disease in people and animals, the information obtained by applying PK/PD principles can be applied reciprocally for the benefit of both, and the environment as well. On the one hand, the use of animal models improves the predictability of preclinical studies used for the development of medicines for human use. On the other hand, quantitative PK/PD information obtained from clinical research can be useful in veterinary drug development. As Schneider et al. describes, in reverse translational pharmacology, PK/PD modeling can help in the selection of therapeutic candidates and the prediction of their optimal dosing schedules (i.e., dose and frequency) both in humans and animals. In addition, this can be done *through extrapolation of disposition kinetics, efficacy, and safety data from/to spontaneous animal models of the human disease pathophysiology to/from the clinic*, because the knowledge obtained from existing resources can be used one way or the other (from humans to animals or *vice versa*) [10]. 

In this review, we present an overview of the PK/PD principles and the models to evaluate the relationship between the PK and the PD of antibiotics, including breakpoints establishment, susceptibility surveillance, therapeutic drug monitoring, and prediction of resistances.

## 2. Pharmacokinetic/Pharmacodynamic Principles

The major indicator of the effect of the antibiotics (PD) is the minimum inhibitory concentration (MIC), which provides information about the susceptibility of the pathogen against the antibiotic [11]. However, the clinical outcome is conditioned not only by the MIC value, it depends on the interactions among the host, the bacteria responsible for the infection, and the administered antibiotic. The PK/PD analysis allows integrating and analyzing jointly both the PK and PD information for drug use optimization.

The combination of pharmacokinetic and pharmacodynamic parameters can be defined through the PK/PD indices. Depending on the activity pattern of the antibiotic, three PK/PD indices have been established as the best descriptors of clinical efficacy and bacterial kill characteristics of the antibiotic. The first pattern of antimicrobial activity exhibits concentration-dependent activity and the PKPD indices preferred are the ratio of the free-drug maximum concentration (*f*Cmax) to the MIC (*f*Cmax/MIC) or the area under the free-drug concentration–time curve, typically over a 24-h period, to the MIC (*f*AUC_24_/MIC). For the time-dependent pattern, the antibacterial effect is best described by the percentage of time the free drug concentration remains above the MIC throughout the dosing interval (*f*T_>MIC_). Finally, the best PK/PD ratio for concentration-dependent with time-dependence antibiotics, is *f*AUC_24_/MIC [5,9,12,13]. The PK/PD indices have also been related to suppression of emergence of resistance [14]. Table 1 summarizes the PK/PD indices related to the efficacy of different antimicrobials.

Once the activity pattern is known, the magnitude of the index that is required for antimicrobial efficacy should be established. This cut-off value, known as the pharmacodynamic target (PDT), is the magnitude for a PK/PD index at which a desired level of predicted response is achieved. During the development of new antimicrobial agents, the magnitude of the PK/PD index usually derives from studies in animals and/or in vitro PD studies, although in more advanced stages of drug development, the results of clinical PK/PD–response studies should be considered in conjunction with results from pre-clinical PK/PD studies.

In the PK/PD analysis, the probability to reach the targeted exposure can be estimated from the PK parameters of the antibiotic in a population. When the inter-patient variability is included in the PK/PD analysis, the Monte Carlo simulation is a useful approach. Monte Carlo methods are stochastic computational algorithms based on repeated random sampling. To carry out the PK/PD analysis by Monte Carlo simulation, two elements are needed: (i) A validated PK/PD model that provides the population PK parameters, the inter-individual variability, and a covariate model that provides information of the influence of patient characteristics on the PK parameter, and (ii) the PD model that integrates both the PK and PD parameters [9]. 

The probability that a specific value of the PK/PD index associated with the efficacy of the antibiotic is achieved at a certain MIC is defined as the probability of target attainment (PTA). The PTA corresponds to the percentage of simulated patients with an estimated PK/PD index equal to or higher than the value related to the efficacy of the antibiotic against a pathogen with a certain MIC. Potentially efficacious dose regimens must provide a PTA > 90% [11,15]. The expected population PTA given a population of microorganisms for a specific dosing regimen is named cumulative fraction of response (CFR) [16]. The CFR can be considered as the expected probability of success of a dosing regimen against bacteria in the absence of the specific value of MIC, and thus, the population distribution of MICs is used.

## 3. Models to Study the PK/PD of Antimicrobials

To assure the quality of drugs, the European Medicines Agency (EMA) and its Committee for Medicinal Products for Human Use (CHMP) provide guidelines for the evaluation of medicinal products, which in the area of antibacterial agents include microbiological research, studies in animals, PK/PD investigations, and clinical trials [17,18]. The main objective of PK/PD studies is to establish the relationship between the PK/PD indices, and the clinical or the microbiological outcomes in patients, and one of the most important applications is regimen selection for clinical studies [19]. Investigation in PK/PD of antimicrobials can be considered as an iterative process through which in vitro and in vivo experiments, population PK models, and in silico simulations are used to evaluate potential dosing regimens and PK/PD targets. Each one displays strengths and weaknesses and may be regarded as complementary. The selection of the adequate analysis is critical for the determination of the dosage regimens. 

Preclinical models require robust indicators for antibacterial activity in humans, and therefore, their results are very useful to optimize dose regimens for efficacy and prevention of resistance [5]. The EMA guidelines on the use of the PK and PD in the development of antimicrobial medicinal products [17] recommends testing about 4–5 organisms of the major target species or organism groups. The organisms should be representative of those most relevant to the intended clinical uses and should exhibit MICs of the test agent that include values at the upper end of the wild-type distribution. The PK/PD model methods fell into three categories: In vitro, ex vivo, and in vivo [20,21], and they must be regarded as complementary. A description of these methods, with the main advantages and disadvantages is included in this section.

### 3.1. In Vitro Models

Although the MIC is the major indicator of the effect of the antibiotics it only provides approximate information about the antibacterial effect of the antibiotics [11]. In vitro models go further than the simple assessment of the MIC of the antibiotic against a certain microorganism. They provide detailed information about the magnitude and time course of anti-infective activity in a controlled environment. However, they require a more sophisticated data analysis to capture the observed kill curve profiles to allow for translation. In vitro models may be used to [17]:-Describe the PK/PD relationships for representative organisms and a range of inocula;-Assess the effects of different PK profiles;-Study the relationships between rates of emergent resistance, drug exposure, and duration of therapy.

In vitro results generally correlate well with microbiological efficacy in humans if there is a similar level of bacterial growth and exposure to the antimicrobial as it happens at the in vivo site of infection [22]. However, despite the fact that in vitro studies may be efficient, the translation of in vitro results to the clinic may be hampered. Some reasons are related to the lack of in vivo physiological processes (for instance, the lack of a functioning immune system) and others, since not all conditions for antibiotics and bacteria in vivo may be known.

#### 3.1.1. Static Assays

The simplest model for in vitro PK/PD studies consists of adding the antibiotic in a constant concentration to the culture medium with a certain number of bacteria, and evaluating the change in bacterial count at different times. This model can be useful to assess the exposure-response relationship against the predominant bacterial population of a single antibiotic, and to evaluate PD drug interactions of antibiotic combinations [20]. The concentration range of the antibiotic, the composition of the culture medium, and the initial size of the bacterial inoculum are key factors [23]. The concentration range must include an antibacterial agent’s concentration that does not kill any bacteria, as well as a concentration that completely kills the bacteria. The culture medium must ensure the growth of the bacteria, and accurately simulate the conditions in vivo. The most important advantage of static models includes the simplicity, low cost, and minimal equipment requirements. As main disadvantages, possible antibiotic degradation, constant exposition of the bacteria to the antibiotic, and short duration of the experiments (typically 24–48 h) can me mentioned [20].

#### 3.1.2. Dynamic Assays

In the dynamic models, the concentration of the antimicrobial changes over time simulating the concentration versus time profile that happens in the patient [24]. They are classified into two categories depending on the number or compartments: One-compartment and two-compartment models. There are also more specific models useful to evaluate specific infections, such as the bladder infection models that will also be described below.

One-compartment model

In this model, also named chemostat, the antimicrobial agent is added to a culture bioreactor containing the bacterial inoculum, where the antibiotic is also administered [24]. The culture medium is continuously removed and replaced by a fresh medium to maintain the total volume as constant (generally about 100–250 mL). The system can simulate the time course of the antibiotic concentration as it occurs in vivo in a patient. Different administration strategies may be assayed: Bolus, infusion or first order input. As in the static model, the change in bacterial count at different times is measured. Experiments normally last up to over 96 h, but frequently they are conducted in shorter times, over 24 h. Figure 1 represents the different modalities of the system depending on the administration type.

This model has shown to be useful in evaluating bacterial killing and regrowth with antibiotics in monotherapy, and combinations as well. In a recent study, the efficacy of ceftolozane-tazobactam in combination with colistin against extensively drug-resistant *Pseudomonas aeruginosa*, including high-risk clones, was evaluated using the chemostat system [25]. The authors observed an additive or synergistic interaction for ceftolozane -tazobactam with colistin, and particularly against resistant strains. The chemostat system has some limitations, including elimination of bacteria, contamination of the medium, incomplete oxygenation, and accumulation of waste products over time [20].

Two-compartment model

In two-compartment PK/PD models, contrary to one-compartment ones, bacterial washout is prevented by physically separating bacteria from the central reservoir within a small peripheral compartment (typically 10–20 mL) [24]. The most common example is the hollow fiber infection model (HFIM). The system consists of thousands of small tubular fibers (filters) in a cartridge through which the medium is pumped from the central reservoir. The wall of the fibers has pores that retain the microorganisms, but allows the free diffusion of the antibiotic. Therefore, while the bacteria are entrapped in the extracapillary space of the hollow fiber cartridge, the fresh medium and antibiotic diffuse through the fiber wall to the extracapillary space (Figure 2). 

As with the chemostat, the HFIM allows simulation of a range of drug disposition profiles and bacteria loads. A major advantage over other in vitro methods is that the duration of the experiments is virtually unlimited [26], and durations up to several weeks have been published for *Mycobacterium tuberculosis* [21,27]. As experiments may last long periods, this method allows evaluating the relationship between drug exposure and resistance development.

One of the most important limitations of HFIM is the high cost, mainly due to the price of the cartridges. Another drawback is the binding of lipophilic antibiotics to the components of the hollow fibers, although different materials are available to minimize binding [20]. 

In order to design and work with one of these in the vitro model, it is important to take into account a series of considerations covering the selection of the most adequate strain, including reference strains, the initial bacterial inoculum, the duration of the therapy and resistance prevention, the stability of the antimicrobial, the antibiotic concentration profile. It is also important to have adequate methods to quantify the drug concentration and bacterial populations [20].

#### 3.1.3. In Vitro PK/PD Bladder Infection Models

To investigate uncomplicated lower urinary tract infections (UTIs), normal urodynamics must be considered. Experimental models must take into account dilution of bacteria during bladder filling and loss through voiding. Over the past years, different models have been developed [28]. In a recent study, a new multicompartment infection model applies a continuous elimination into 16 bladder compartments [29]. This model has been used to evaluate the oral fosfomycin treatment for enterococcal urinary tract infections. A next-generation “micromodel” has been recently developed and it has been used to analyze the impact of urinary flow on the persistence of *E. coli* colonization [30]. This dynamic in vitro model used transitional epithelial cells and type IV collagen, and by simulating urinary tract shear stresses and flow velocities, the dynamics of *E. coli* cell adhesion was studied. Using this model, the authors reported a phenomenon of epithelial cell “rolling-shedding” that promotes bacterial attachment into deeper layers of epithelial cells.

In vitro UTI models try to mimic as closely as possible the conditions at the site of infection. However, an important drawback limits the translation to humans. For example, although the urinary bladder contains relatively low oxygen (PO_2_ is about 40 mmHg), in vitro models are generally run at normal atmospheric conditions [28].

### 3.2. Ex Vivo Models

PK/PD relationships can be studied using ex vivo models, such as the tissue cage (TC) model. This model consists of perforated cylinder, tubes or spheres, implanted in subcutaneous tissues. After implantation (3 to 4 weeks), the granulation tissue surrounds and partially fills the cage, the remainder being filled with an interstitial fluid which can be inflamed with carrageenan (to produce a sterile exudate), infected (inoculated septic exudate) or not (transudate), allowing antimicrobial action to be monitored as a local, isolated infection [21]. TCs are, actually, test tubes implanted into the animal, enabling ethically accepted sequential sampling. The effect of the antibiotic can be directly assayed by determining the concentration of the drug and its metabolites (active or inactive) and by monitoring bacterial counts in the TC fluid. The ex vivo killing effect of the antimicrobials in the collected fluid (transudate, inflammatory exudate) may be evaluated through time-kill curve assays [31]. 

In a recent study, the potential synergistic interaction between tigecycline and aminoglycosides against clinical isolates of carbapenem-resistant *Klebsiella pneumoniae* (CR-KP) was assessed using a tissue-cage infection model of rats [32]. The study revealed that, compared with single drugs, tigecycline combined with aminoglycosides could exert synergistic effects and reduce the emergence of tigecycline resistance. Therefore, the study allowed the conclusion that this combination might be an effective alternative when treating CR-KP infections in clinical practice.

One advantage of the TC model is the presence of natural immunity. Moreover, it allows detecting matrix-specific effects of drug action. For instance, thymidine concentration in serum is high in cattle, rats, and mice, but low in dogs and man. This compound is a known antagonist of the action of trimethoprim on some bacteria, such as *E. coli*. In a previous study using a TC model in calves, the authors unexpectedly did not detect the effect of trimethoprim on *E. coli*, since the high levels of thymidine were enough to antagonize the antibacterial effect of the antimicrobial [33]. These findings show the inconvenience of extrapolating data from one species to the other.

### 3.3. In Vivo Animal Models

Various established infection animal models have been used for the experimental antibacterial PK/PD evaluation. The advantage of the animal models is that they can truly reflect the progress of drug and bacterial exposure in the animal model. The effect of virulence, immune function, the injection of the test microorganism, and the antibiotic concentration at the infection site are the main determining factors. The most important disadvantage of the in vivo models is that the drug concentration cannot be quickly measured [23].

Zhao et al. [34] have reviewed animal infection models that have been extensively used in antibiotic discovery and development, including PK/PD analyses. These models involve the inoculation to the animal of a certain number of bacteria. In general, mice and rats are the preferred experimental animals due to their low cost and ease of handling. To induce the infection, virulent bacterial strains are used, and to produce a progressive infection, high inoculums, immunocompromised animals, and/or adjuvants (formalin or mucin) may be required. The primary end-points is the reduction of bacterial burden in the infected tissue (expressed as colony-forming units (CFU)), which is typically assessed at 24 h after initiation of the antimicrobial therapy. According to the CHMP guidelines, pathogen specific PK/PD targets resulting in a net stasis, a 1-log_10_, and 2-log_10_ reduction of CFU must be provided [15].

The most relevant animal models of PK/PD studies are the murine thigh and lung infections. Others include skin and soft tissue infections, septicemia, meningitis, urinary tract infections, endocarditis or intraperitoneal infections [34]. In most studies, neutropenia is induced with cyclophoshamide to minimize the effect of the immune system as a confounder. Models of infection for specific microorganisms have also been reported. For example, a recent publication describes in vivo mouse models of both local and systemic *Staphylococcus aureus* infection, and presents protocols for models of subcutaneous infection, tape stripping skin infection, sample collection to determine skin structure, production of inflammatory mediators, and bacterial load, post-traumatic osteomyelitis model, and intravenous infection of the retro-orbital sinus [35]. 

In the thigh infection model, an inoculum of 10^5−8^ CFU of a certain pathogen is injected intra-muscularly, and after therapy with the antibiotic, the animal is euthanized. The homogenized tissue and the reduction in bacterial burden in the mice’s thighs indicates the efficacy [34]. This model is generally used in the development of new antibiotics, and it has been shown to be helpful for predicting efficacy for a number of indications, including pneumonia, skin and soft tissue infections, intra-abdominal infections or septicemia [34]. One advantage is that the two thighs of the animals can be used, allowing a reduction in the number of animals per experiment. 

Among others, lung infection models include those that mimic human pneumonia [36]. Pneumonia can be induced in the animals by different methods: Exposure to aerosol, transtracheal injection, peroral intubation or intranasal inoculation, and each has specific strengths and weaknesses. In a recent study, to evaluate the effect of omadacycline against *S. aureus*, neutropenic mice were infected by the intranasal route. Two hours after inoculation into lungs, the mice were treated with the antimicrobial by the subcutaneous route, and 24-h later, the animals were sacrificed and the burden of organisms was enumerated from lung homogenates. The dose-response curves revealed the potent in vivo effect of omadacycline against 10 MSSA (methicillin-susceptible *S. aureus*)/MRSA (methicillin-resistant *S. aureus*) strains [37].

Biofilm-related animal models have also been developed with or without the addition of foreign material, including central venous catheter models, subcutaneous foreign body infection models, and osteomyelitis infection models. The microorganisms inoculated are usually planktonic but capable of attaching to surfaces and developing biofilm [38]. Dalton et al. [39] developed an in vivo polymicrobial biofilm wound infection model to study interspecies interactions in biofilms and their relation to wound chronicity.

For real-time monitoring of infections, animal models using luminescent bacteria have been developed. These models allow following the course of infection in live animals in a non-invasive manner. However, as compared with viable counting methods, the sensitivity of bioluminescence is generally lower. Consequently, they are not useful to evaluate compounds with mild antibacterial effect [40].

## 4. PK/PD Modeling of Microbial Kill-Curves

Mathematical modeling to analyze PK/PD data resulting from in vitro, ex vivo or in vivo studies has an important impact on the development and optimization of antibiotic dosing. The complexity of the PK/PD models depends on the type of data generated from the experiments. The kill-curves, where a time course of drug-bacterial response is produced, have been used to describe bacterial growth and death rates, drug effects, and the emergence of resistant strains within a population [41]. An example of a time kill-curve profile is shown in Figure 3.

### 4.1. MIC-Based Approach

Classification of antibiotics into time-dependent versus concentration-dependent killing has guided the dosing of antimicrobials for many years. This was achieved through relating drug exposure to the minimum inhibition concentration (MIC). The estimation of the best PK/PD index of a certain microorganism is calculated by plotting the value of an efficacy endpoint (typically log_10_ CFU/mL after 24 h) versus the magnitude of each of the three PK/PD indices [42] (Figure 4). Although the application of MIC-based PK/PD indices may be useful for optimization of dosing regiments, it is actually, a simplification of the PK/PD relationship.

### 4.2. Mechanism-Based Models

To analyze the PK/PD data, mechanism-based models can be applied. These models consider the two most important factors that determine how effective a treatment will be for a given patient: The free and available antibiotic concentration at the infection site (PK), and the susceptibility of the bacteria (PD) [43]. These models describe the time course of drug effects and disease, and implement the mechanism of action of the antimicrobial or the mechanism of resistance, thereby maximizing the information gained from experimental data. They can simultaneously describe and predict the time course of bacterial killing and resistant emergence and can be applied to evaluate the effect of antimicrobials in monotherapy and in combination [20]. 

The mechanistic models consider several parameters rather than the simple MIC value. They usually include, at minimum, a control growth rate constant (K_growth_) and a killing rate K_death_), a maximum kill rate (E_max_) and a potency value such as the half-maximum effect concentration (EC_50_) [44]. Some authors have shown a relationship between the MIC and the EC_50_ for a certain antibiotic/bacteria combination [45]. However, since the MIC is defined by the absence of visible growth at a certain time point, it is not conditioned by the maximum kill rate, and therefore, provides less information. In fact, similar values of MIC can be obtained with different growth and kill rates [44]. Over time, mathematical models that describe the microbial kill-curves have become more detailed and sophisticated. For instance, more complicated models include delay functions or factors to evaluate the combination of antibiotics [46]. The Figure 5 shows a schematic representation of a model used to characterize the aztreonam-avibactam killing effect on drug susceptible and less-susceptible bacteria, designed from data obtained with a static-kill model [47].

## 5. Application of PK/PD Analysis and Population Pharmacokinetics for the Development and Optimization of Antimicrobial Treatments

PK/PD modeling implies the use of mathematical equations to express drug-related biological changes, what makes it a useful tool to describe the kinetic and dynamic relation for new drugs, to determine the optimal dosing regimens, but also to establish susceptibility breakpoints, and to optimize pre-existing dosing regimens with direct application on clinical situations and in veterinary medicine [9].

### 5.1. PK/PD Analysis in Drug Development 

The application of PK/PD principles is a powerful tool to elucidate the clinical potential of antimicrobial agents in early stages of development [48]. Nowadays, different sources of PK/PD data are integrated during the drug development procedure, which provide a stronger evidence to support the approval of new antimicrobials [49]. Animal PK/PD studies and in vitro models have relevant influence on indications and recommendation of dosing regimens. Additionally, thanks to preclinical PK/PD studies, smaller sample sizes may be used in clinical trials [5]. As a result, both the EMA and the FDA consider PK/PD experiments and simulations as a key step in the development process of antimicrobial agents [5,15,17,18,49,50]. Actually, data from in vitro and animal infection models are required for entering first-in-human studies, and after clinical development, to obtain market approval by the two regulatory agencies. For instance, for the approval of dalbavancin, the in vivo PK/PD relationship for *Staphylococcus aureus* was investigated using a neutropenic model of animal infection that showed that net reduction in the log_10_ of colony-forming units was greatest when larger doses were given less frequently [51].

One of the main challenges of the development of an antimicrobial drug is to determine the correlation between in vitro susceptibility and clinical efficacy. An integrated model-based approach plays a relevant role in drug development and evaluation, since it enables an informed decision-making process at the preclinical and clinical development stages [52]. As discussed above, validated in vitro PK/PD and animal infection models have been extensively used for identifying the most predictive PK/PD indices, offering a support to optimize study designs in terms of minimizing the cost and duration of clinical trials, increasing the success rates, and accelerating the drug development process. These preclinical PK/PD models are a key point in the search for new antimicrobial treatments [24]. 

The use of PK/PD analysis has led to a reduction or sometimes, to a replacement of clinical dose-finding studies during the clinical development of new antimicrobial agents. The EMA defends that the PTA can be used to predict whether a treatment may be useful against specific microorganisms, and underlines the relevance of this fact for infections caused by multi-resistant and rare bacteria, since *very few are likely to have been treated in pre-licensure clinical studies and it may be very difficult to interpret the clinical outcome data* [15]. 

In any case, after the preclinical evaluations, the human PK information is critical for the selection of potentially effective dose regimens. Population PK models have to be developed in order to predict human exposures to the drug and to explore exposure-response relationships in the target population [15]. The population PK approach during drug development allows integrating information of pharmacokinetics from sparse, dense or a combination of sparse and dense data. The results obtained from data-rich Phase I studies, can be used to establish or optimize PK and PK/PD models and to quantify for the first time the variability in humans [52]. From these early phases until drug commercialization, the model based drug development allows using knowledge at every stage, integrating all the information, which makes it possible to refine the models iteratively and optimize decision making. 

In order to obtain reliable results evaluating dosing regimens, the development of population PK models is imperative. The use of population approach has been one of the major developments in pharmacometrics. Population PK models help define the sources and correlates of pharmacokinetic variability in target patient populations and their impact upon drug disposition [53]. The main objective of population PK models is to determine the mean values of PK parameters (i.e., clearance, volume of distribution), inter-individual variability, intra-individual variability, and between-occasion variability, and to account for explanatory covariates of interest [54]. Population PK allows identifying demographical, physio-pathological, therapeutical or other features that vary between subjects, and that could be responsible for some of the differences in the achieved drug concentrations. 

A large evolution in the field of data analysis for drug development has been observed and, concurrently, the terminology has also evolved. The term Model Informed Drug Discovery and Development (MiD3) was firstly defined in 2016 [55] as *quantitative framework for prediction and extrapolation, centered on knowledge and inference generated from integrated models of compound, mechanism and disease level data and aimed at improving the quality, efficiency and cost effectiveness of decision making*. The MiD3 workgroup considers this as a *holistic term to term to characterize a variety of quantitative approaches used to improve the quality, efficiency, and cost-effectiveness of decision making through ‘‘fit-for-purpose’’ data analysis and prediction*. Figure 6 depicts the evolution in the approaches and terminology in this field. These approaches have a long history in the field of infectious diseases and their application is closely linked to antimicrobial drug development [44]. Antimicrobial agents’ development will require the integration and connection of different approaches and the use of translational tools to support decision making. Moreover, Rayner et al. in a recent review suggest the incorporation of deep learning and artificial intelligence approaches [44]. 

During drug development, the PK/PD analysis can also help in the design of the dosage forms, since they strongly determine the efficacy of antimicrobial agents. For instance, the PK/PD targets in antibiotics with time-dependent activity and short half-life, such as beta-lactams, could be addressed with controlled-release dosage forms. In 2004, the pharmacokinetically enhanced amoxicillin/clavulanate (2000/125 mg) twice daily formulation demonstrated higher T_>MIC_ values and was more effective than immediate release formulations, achieving substantially better eradication rates against *S. pneumoniae* isolates with MICs commonly encountered in the clinic [56]. More recently, Li et al. [57] evaluated a three-pulse release tablet for amoxicillin using a physiologically based pharmacokinetic modeling (PBPK) and they also concluded that their formulation extended the effective plasma concentration compared to the immediate release tablet. In veterinary medicine, with the same aim, Horwitz et al. developed a novel gastroretentive controlled-release drug delivery system for amoxicillin therapy [58].

More extensively, the application of biopharmaceutical pharmacometrics (BPMX)—i.e., pharmacometric modeling incorporating biopharmaceutical principles has been demonstrated as a valuable tool in drug discovery and development. For instance, Sou et al. [59] reported the application of BPMX for the PK analysis of three investigational antimicrobial agents following pulmonary administration of different suspension formulations. They developed a PK model considering formulation properties and provided a mechanism to estimate dissolved drug concentrations in the lungs. The model was able to predict that these antibiotics for lung delivery should ideally be delivered in a sustained release formulation with high solubility for maximum local exposure in lungs for efficacy, with rapid systemic clearance in plasma for reduced risk of unwanted systemic adverse effects.

### 5.2. PK/PD Analysis in Setting Susceptibility Breakpoints

An antimicrobial breakpoint is an established concentration value which essentially defines at what value a microorganism is considered susceptible, intermediate or resistant to antimicrobial therapy, and therefore is used to guide clinicians on the prescription of antimicrobial drugs in the clinical practice [9,60]. The PK/PD analysis and Monte Carlo simulations provide a useful platform for the establishment of breakpoints based on the likelihood of obtaining a targeted exposure [61,62,63]. In the early 2000s, different studies started, applying PK/PD simulations in the establishment of antimicrobial breakpoints and comparing the results with The Clinical and Laboratory Standards Institute (CLSI) and European Committee on Antimicrobial Susceptibility Testing (EUCAST) breakpoints in force at the time [61,64,65,66]. More recently, the PK/PD criteria have been used to evaluate the susceptibility breakpoints of multidrug resistant bacteria such as *Mycobacterium tuberculosis* [67,68]. Table 2 includes several examples of published studies that review breakpoints by applying the PK/PD criteria.

Both EUCAST and CLSI have developed tools to apply the PK/PD analysis to set and revise breakpoints. The EUCAST approach about the role of PK/PD in setting clinical MIC breakpoints was reported in 2012 (Figure 7). Contrary to clinical breakpoints, PK/PD breakpoints are regimen-dependent (species-independent) and therefore, different PK/PD breakpoints can be obtained for the same drug [60]. Moreover, due to the increase of PK/PD knowledge in this field, recently, CLSI defined a “susceptible dose-dependent (SDD)” category for certain drug and organism combinations. In the same way, EUCAST proposed a new definition for the intermediate category (I): “Susceptible, increased exposure (I)”. CLSI defines the SDD category as a breakpoint that implies that the susceptibility of an isolate depends on the dosing regimen that is used in the patient. For isolates that are in the SDD category, it is necessary to move to a dosing regimen (i.e., higher doses, more frequent doses or both) that results in higher drug exposure than that achieved with the dose that was used to establish the susceptible breakpoint [69]. EUCAST considers a microorganism as “Susceptible, increased exposure” when there is a high likelihood of therapeutic success when exposure to the agent is increased by adjusting the dosing regimen. Exposure depends on the dosage regimen (route of administration, dose, dosing interval, infusion time), as well as distribution and excretion of the antimicrobial agent, since they condition the drug concentration at the site of infection [15].

### 5.3. PK/PD Analysis as A Tool For Surveillance of Antibacterial Activity

The development and establishment of programs of integrated epidemiologic surveillance on antimicrobial activity is essential to determine risk factors, to identify temporal trends in resistance patterns, and to guide the clinician towards appropriate empiric treatments [70]. Antimicrobial stewardship programs are needed to choose the treatment properly in terms of optimal choice of drug, dosage, and duration of treatment, which leads to a reduction of treatment-related costs, an improvement in clinical outcomes and safety, and a reduction or a stabilization of antimicrobial resistance [71,72,73,74]. Standard surveillance indices, based on MIC values, can be useful but they are not enough to detect changes on the whole activity of antimicrobial agents, since some less evident variations in MIC distribution could lead to a loss of treatment efficacy [71,75]. The PK/PD analysis has proved to be a useful tool to establish empirical antimicrobial therapy recommendations, since it allows evaluating the overall activity of antimicrobial treatments by predicting the probability of success for a treatment, incorporating the variability of the pharmacokinetic parameters and the bacterial population MIC distribution (local MIC distributions).

Accordingly, during the last years, the Survey of Antibiotic Resistance (SOAR) has published periodically the survey results of Antibiotic Resistance in different countries to provide a picture of the state of antibiotic susceptibility of different pathogens. The Survey of Antibiotic Resistance (SOAR) is an international antibiotic resistance surveillance study that focuses on key respiratory pathogens from community-acquired infections and has been running since 2002 in the Middle East, Africa, Latin America, Asia-Pacific countries, and Commonwealth of Independent States countries. It analyzes data based on CLSI, EUCAST (dose-specific), and PK/PD breakpoints [76,77,78,79,80,81,82,83,84,85,86,87,88,89,90,91,92,93,94,95,96]. The study concluded that there are large country-specific differences in antibiotic susceptibility even within the same region and reinforced the need for regular antibiotic resistance surveillance in order to track susceptibility changes over time.

Additionally, many publications include the PK/PD analysis as a key point for the treatment optimization and surveillance of antibacterial activity. As an example, different studies about *P. aeruginosa* isolates conclude that susceptibility rates and the probability of treatment success estimated by the PK/PD analysis are complementary tools for surveillance purposes and both should be considered together when making decisions to guide antimicrobial therapy [71,72,75]. A recently published review about treatment optimization for sexually transmitted infections also highlights the urgent needs of systematic PK/PD evaluation to ensure resistance suppression and bacterial eradication at all sites of infection [97]. Finally, the PK/PD analysis has also been demonstrated to be useful in identifying changes in antimicrobial activity after the implantation of vaccination programs, since complementary information to the susceptibility rates or the MIC values is provided [98,99]. Regular epidemiologic surveillance programs have an important role to guide the clinician towards appropriate empiric treatments and they should incorporate the local ecology to provide a personalized approach. In this regard, the monitorization of the CFR values, allows estimating the probability of success for a treatment without knowledge of the susceptibility of the specific isolate responsible for the infection, but taking into account the MIC distribution of a particular institution or hospital wards or regions/countries. 

### 5.4. Population PK and PK/PD to Optimize Dosing Regimens. Therapeutic Drug Monitoring (TDM)

Many efforts have been made along the last years in order to improve the use of antimicrobials and to reverse AMR. Consequently, different strategies have been implemented in hospital care to optimize the antimicrobial dosing regimens [100] and in this field, the use of PK/PD analysis has become an essential tool to be included in antimicrobial stewardship programs [54]. Moreover, the Guidelines for Developing an Institutional Program to Enhance Antimicrobial Stewardship published by the Infectious Diseases Society of America, and the Society for Healthcare Epidemiology of America, in 2007 already stated that the *optimization of antimicrobial dosing based on individual patient characteristics, causative organism, site of infection, and pharmacokinetic and pharmacodynamic characteristics of the drug is an important part of antimicrobial stewardship* [101]. 

The integration of basic PK/PD relationship, defined from in vitro and animal models, with human PK studies by mathematical modeling and simulation that considers patient variability creates a powerful tool to inform dosing strategies. The statistical method most often used is Monte Carlo simulations (MCS), but other methods may be used. Monte Carlo simulation can be performed using basic PK data which considers mean PK parameter values and variability. However, the better approach to perform Monte Carlo simulation requires: (i) A validated population PK model including the structural model (providing population PK parameters), a variability model (providing inter-individual variability), and a covariate model (studying the influence of patient characteristics on the PK parameters) and (ii) a PD model where the interrelationship of the PK and PD parameters has been studied [9]. Whenever possible, the PK data for simulations should be based on population PK models built from or including PK data from the infected target patient population, with an assessment of the effect of covariates [15]. Using simulations, it is possible to estimate the probability of attaining the target (PTA) over a range of CMI, and the cumulative fraction of response (CFR) when a MIC distribution is available (useful for empiric treatment).

The population PK analysis is a well-accepted approach not only to optimize dosing regimens during new drug developments, but also to improve dosing regimens of old antibiotics and to individualize the antimicrobial treatment in the clinical setting [54]. In fact, optimization of the clinical use of old drugs is of main concern and must be as prioritized as the determination of the therapeutic role of new antibiotics [102]. The covariate analysis in population PK modeling allows evaluating the impact of clinical or demographic parameters on the exposure to the drug and helps in elucidating whether dose adjustments are needed in specific populations. As an example, obese patients, pregnant women, pediatric patients, patients with renal or hepatic impairment or critically ill patients are at high risk of treatment failure or toxicity, as a consequence of physiological or pathological changes that often result in changes in the PK behavior and achievement of PK/PD targets [49]. Antimicrobial dosing in these populations is complex and remains a challenge. Therefore, robust population PK models have to be developed including data from target patient populations with the target indication and considering the adequate PK/PD target.

The physiologically based pharmacokinetic (PBPK) approach is increasingly being used to provide information about drug disposition and exposure in specific subpopulations or clinical situations or to evaluate drug-drug interactions. However, it is essential to keep in mind that PBPK models are based on estimations of some physiological parameters such as blood flow, which can be strongly different in special populations. Therefore, it should be noted that this approach requires robust physiological and PK data in the patient population [49]. Table 3 lists some research works that apply the PBPK approach to optimize antimicrobial use in special populations. These studies show the usefulness of PBPK for prediction of concentrations in different groups of patients, and ultimately for dose individualization in different subgroups of patients.

All in all, the PK modeling methods have been applied to design rational regimens for administration of new and old antibiotics. Population PK models can be used to simulate multiple dosing regimens and to predict the probability of target attainment in different populations. As an example, PK/PD principles have been applied to defend the use of extended and continuous infusion rather than short infusions of beta-lactams in septic or critically ill patients [9,112,113,114,115,116,117]. Beta-lactams exhibit a time-dependent antibacterial activity and therefore extending the duration of perfusion increases the probability to attain the PK/PD target. Recently, continuous infusions have also been proposed to optimize the treatment with linezolid in critically ill patients, especially those with augmented renal clearance [118,119]. For antibiotics with concentration-dependent activity, other authors recommend the daily dose once a day rather than lower doses more times a day, which may reduce drug toxicity [120,121,122].

However, given the high inter-individual variability in certain circumstances where antimicrobial PK is significantly altered (i.e., critically ill, patient with augmented renal clearance, elderly, pregnant women or subjects with extreme body mass index), therapeutic drug monitoring (TDM) may be indicated [123]. TDM is used to individualize dosing with the aim of maximizing the probability of attaining therapeutic success and minimizing the probability of toxicity and development of antimicrobial resistance [124]. Initially, the purpose of TDM was limited to molecules with a narrow therapeutic range such as aminoglycosides or glycopeptides, but it is extending its use to other drugs in specific subgroups of patients or in situations where the bacteria responsible for the infections shows a limited susceptibility to antimicrobial treatments. For instance, although TDM with beta-lactams has not been extensively applied due to their wide therapeutic window, TDM can be useful in relevant populations such as critically ill, obese, burns, neutropenic patients or subjects with renal disease, since it allows determining the dosing regimen that maximizes the probability to attain the PK/PD target and improve clinical outcome [115,124].

Antimicrobial stewardship has been defined as coordinated efforts to ensure that patients receive the right antimicrobial agent, at the right dose, and for the right duration while minimizing adverse drug events and antimicrobial resistance [125]. The first guidelines for conducting antimicrobial stewardship in the hospitalized setting were published in 2007. These guidelines recommend that the stewardship program employs PK/PD principles as well as adopts computerized decision support technologies when possible [98]. In fact, the Infectious Diseases Society of America (IDSA) and the Society for Healthcare Epidemiology of America (SHE) guidelines on antimicrobial stewardship recommend the use of PK/PD dose optimization, as well as the adoption of computer decision supports (CDS) when possible [126]. 

Despite the relevance of PK/PD modeling and simulation programs, they are generally absent in clinical practice for TDM. To facilitate the application of PK/PD in clinical routine, different technologies have been developed. These tools may be specifically used to assist antimicrobial treatment selection and/or dosing regimen design using PK/PD principles and/or Bayesian modeling. Some examples are PK/PD compass (http://www.pkpdcompass.com/, accessed on 1 June 2021), TDMx (http://www.tdmx.eu, Accessed on 1 June 2021), DoseMERx (http://doseme-rx.com, accessed on 1 June 2021), InsightRx (http://insight-rx.com, accessed on 1 June 2021), AMKnom (http://shiny.cumulo.usal.es/amknom/, accessed on 1 June 2021), and MeroRisk Calculator (https://www.bcp.fu-berlin.de/en/pharmazie/faecher/klinische_pharmazie/arbeitsgruppe_kloft/forschung/MRc/index.html, accessed on 1 June 2021). These technologies, web or mobile-based, use population PK models and Monte Carlo simulations to perform the PK/PD target attainment analysis and inform dosing regimens of a variety of antibiotics [98].

### 5.5. Application of PK/PD Modeling to Drug Resistance Prediction

Antimicrobial resistance is an ecological problem that affects the health of humans, animals, and the environment [8]. Taking into account the great threat that the AMR is and the scarce number of new antibiotics, the optimization of the use of currently available antibacterial agents should also consider the minimization of emergence of resistant mutants. The selection of dosage regimens with the aim to suppress resistance can also be guided by the PK/PD criteria. PK/PD targets required to attain this objective have been described in the in vitro and pre-clinical in vivo studies. 

The PK/PD indices related to suppression of emergence of resistance include the conventional PK/PD ratios explained in point 2 but additionally, different works suggest that they should be based on the mutant prevention concentration (MPC). The MPC has been defined as the MIC of the least susceptible single-step mutant. When the antibiotic concentration exceeds the MPC, the cell growth requires an organism to have developed two or more resistance-causing spontaneous chromosomal point mutations [127,128,129]. The concentration range between MIC and MPC is known as the mutant selection window (MSW) (Figure 8), and the selection of resistant bacterial subpopulations is promoted when concentrations are maintained inside this range.

Unfortunately, the target exposures required to reduce the development of resistance are not well defined in many cases. Sumi et al. [14] published in 2019 a systematic review with the aim to describe the currently known antibiotic PK/PD indices required to suppress the emergence of Gram-negative bacterial antibiotic resistance, but further research would be welcome.

Analyzing the available information, the first conclusion is that the exposure required is usually much higher than that to assess the treatment efficacy and therefore, standard dosing regimens are too low to reach the established targets to prevent the emergence of resistance mechanisms [130]. Moreover, the benefits of implementing the high PK/PD targets needed to avoid resistance must be balanced with the potential risks of toxicity. Consequently, the use of alternative dosing strategies, such as the extended or continuous infusions should be considered to enhance the probability of reaching the targeted PK/PD indices for suppression of resistance [14].

### 5.6. Application of PK/PD Modeling in Veterinary Medicine

Antimicrobial agents are the most frequently used drugs in veterinary medicine [23], being the antimicrobial consumption by animals is twice than that used by humans [131]. There is no doubt about their usefulness to prevent disease and to promote growth in food animals, providing healthy animal-source food [132]. However, the abuse and the misuse of antimicrobials in veterinary medicine have led to higher rates of treatment failure and to the development and dissemination of bacterial resistance [23]. Therefore, all efforts have to be undertaken to reduce the consumption of antimicrobials in veterinary and to use them reasonably, and the same principles deserve to be applied in veterinary medicine as well as in human medicine. At this point, PK/PD modeling is an important tool that undoubtedly helps in optimizing the dosing regimens of antimicrobial agents used in veterinary medicine [23,133].

The PK/PD approach was firstly applied in veterinary medicine in the 1990s [21,134,135]. Since then, many research studies have been published PK/PD in the veterinary field. As an example, Table 4 resumes the papers published during the last 5 years. As for humans, in vitro, in vivo, and ex vivo studies are necessary to evaluate antimicrobial drugs for veterinary use. 

In 2002, and in accordance with the existing evidence, the EMA Committee for Medicinal Products for Veterinary Use (CVMP) established in their guideline for the demonstration of efficacy for veterinary medicinal products containing antimicrobial substances, that the PK/PD analysis should be carried out in order to select an appropriate strategy of administration, to optimize dosage, to achieve optimal efficacy, and to minimize the development of resistance [149]. However, in 2004, Lees et al. [21,150] proposed the introduction of the PK/PD approach to setting dosing regimens for veterinary antimicrobials. Later, in 2016, EMA updated the CVMP guideline for the demonstration of efficacy for veterinary medicinal products containing antimicrobial substances to introduce PK/PD studies for dosing selection and to establish breakpoints [151]. Moreover, the application of the EUCAST approach and Monte Carlo simulation to support the selection of a clinical breakpoint is also acceptable. Recently, the CVMP has published the reflection paper on dose optimization of established veterinary antibiotics in the context of a summary of product characteristics harmonization [152], which defends the importance of revising the dosing regimens of older antibiotics due to the risk to increase antimicrobial resistance rates as a consequence of inappropriate exposure. PK/PD modeling should be the tool for decision-making and both the PK and PD variability should be taken into account.

In essence, the application of PK/PD principles in veterinary medicine is widely accepted, although further research is required for a better understanding of the PK/PD relationships of veterinary drugs. New modeling approaches should contribute to improve the antimicrobial use in animals with the aim to avoid unnecessary exposure to antimicrobials, to ensure the efficacy of the treatment, and also to prevent the selection of resistant bacteria. 

## 6. Conclusions

In conclusion, this review outlines the current state of the art of the PK/PD analysis in the development and evaluation of antimicrobials. Regulatory guidelines include the PK/PD approach to advance in effective drug development and regulatory evaluation, not only in human medicine but also in veterinary. Different in vitro, ex vivo, and in vivo methods have been developed to establish the relationship between the PK/PD indices and the clinical or the microbiological outcomes in patients. We highlight the relevant role of this approach in the optimization of existing and new antibiotics, and also in the determination of susceptibility breakpoints and in surveillance programs. Despite the important progress of PK/PD in the last years, some limitations are still present. Thus, additional work is necessary to improve the quality and external validation of in vitro and in vivo studies, and new standardized methodological approaches must be developed. Optimizing the available translational PK/PD tools is also very important to predict successful treatment regimens, and to prevent the development of multi-drug resistant bacteria. In this sense, an important challenge of PK/PD is the application in clinical practice for dose optimization and to prevent the development of multi-drug resistant bacteria. Nevertheless, we have to keep in mind that PK/PD modeling and simulation is continuously evolving and new methods are appearing. In fact, several challenges are needed to be further addressed, such as the management of antibiotic combinations, the relationship or correlation between free drug concentrations in plasma and in the infection site or the development of approaches to restrict the selection of resistant mutants. Consequently, the strategies currently available will change and more complex models with higher ability to predict clinical efficacy and/or emergence of resistances will be developed and implemented.

## Figures and Tables

**Figure 1 pharmaceutics-13-00833-f001:**
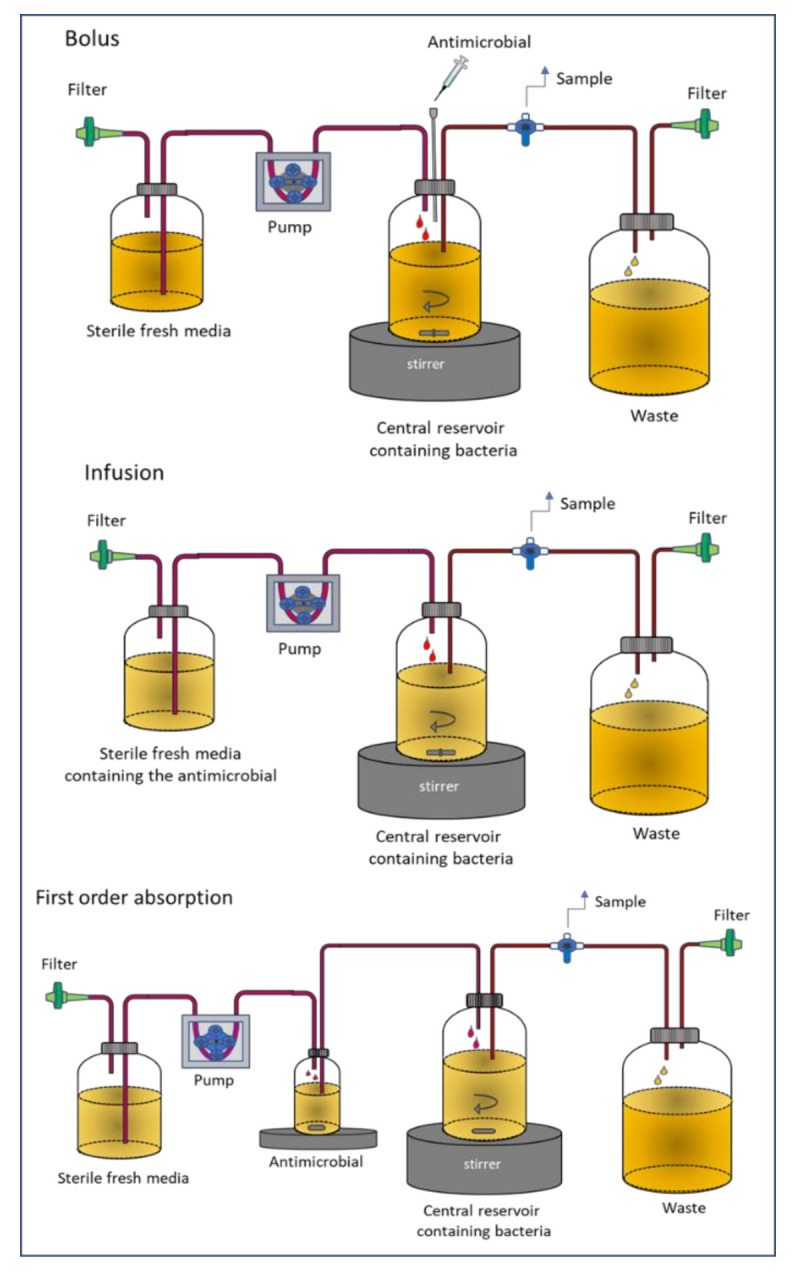
Dynamic one-compartment in vitro infection model (chemostat).

**Figure 2 pharmaceutics-13-00833-f002:**
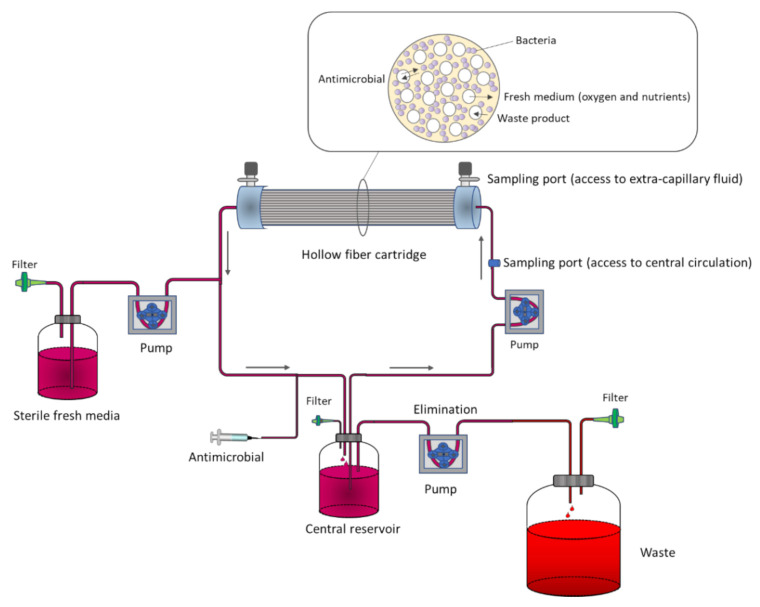
Representation of the hollow fiber infection model (HFIM).

**Figure 3 pharmaceutics-13-00833-f003:**
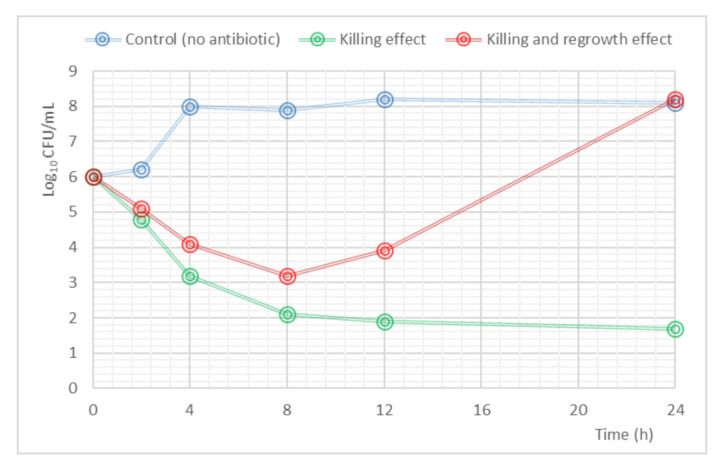
Example of a time-kill curve. The blue line shows the evolution of the initial inoculum in the absence of antibiotic (control); the green line represents the reduction in the bacterial count in the presence of antibiotic (killing effect); the red line explains the behavior in the bacterial load when a regrowth effect appears.

**Figure 4 pharmaceutics-13-00833-f004:**
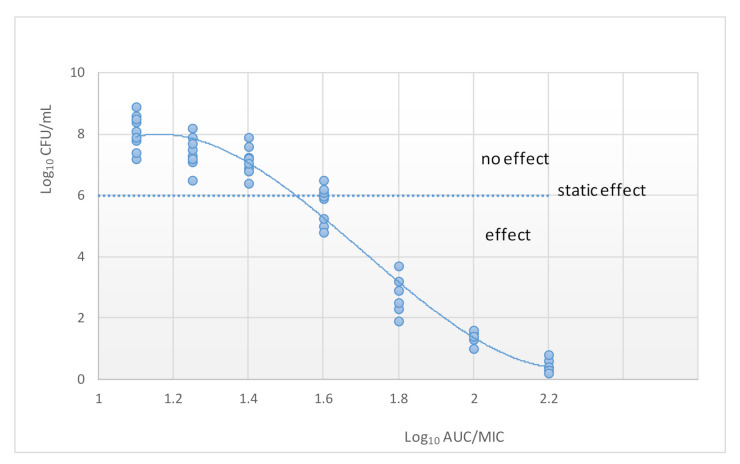
Relationship between the AUC and the number of colony-forming units (CUF). The dotted line indicates the initial inoculum.

**Figure 5 pharmaceutics-13-00833-f005:**
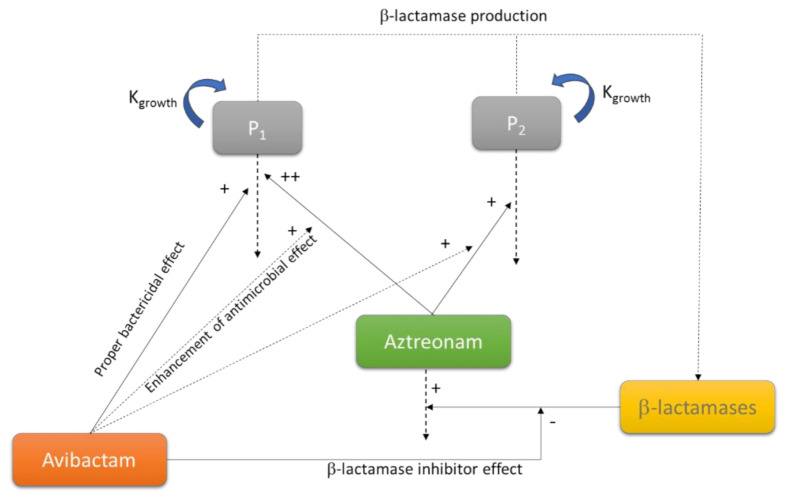
Schematic representation of a model used to characterize the aztreonam-avibactam killing effect on drug susceptible (P1) and less susceptible (P2) bacteria. Adapted from [47]; Published by the American Society for Clinical Pharmacology and Therapeutics, 2019.

**Figure 6 pharmaceutics-13-00833-f006:**
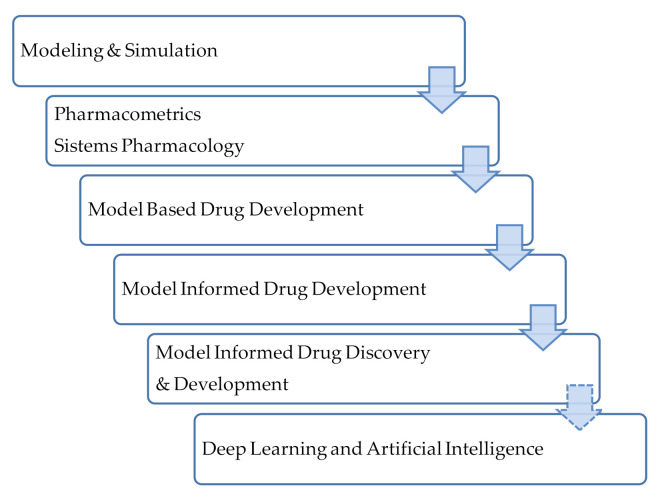
Terminology used for quantitative approaches for data analysis. Adapted from [55]; Published by the American Society for Clinical Pharmacology and Therapeutics, 2016.

**Figure 7 pharmaceutics-13-00833-f007:**
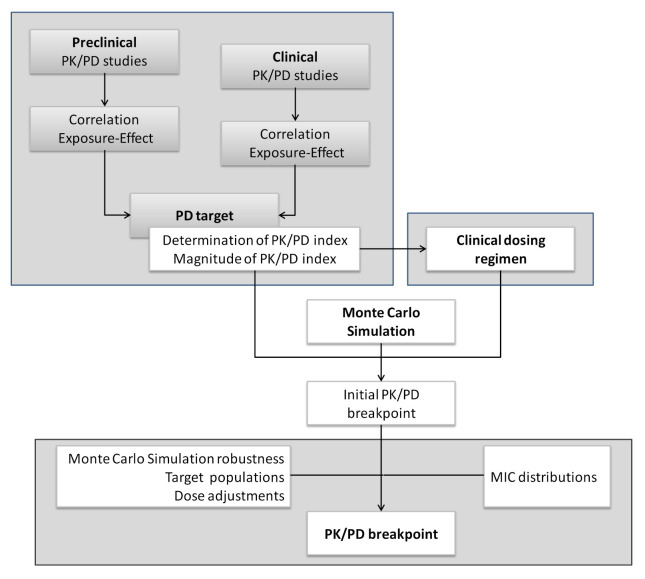
Process of setting PK/PD breakpoints by EUCAST. Adapted with permission from [60]; Published by Elsevier, 2012.

**Figure 8 pharmaceutics-13-00833-f008:**
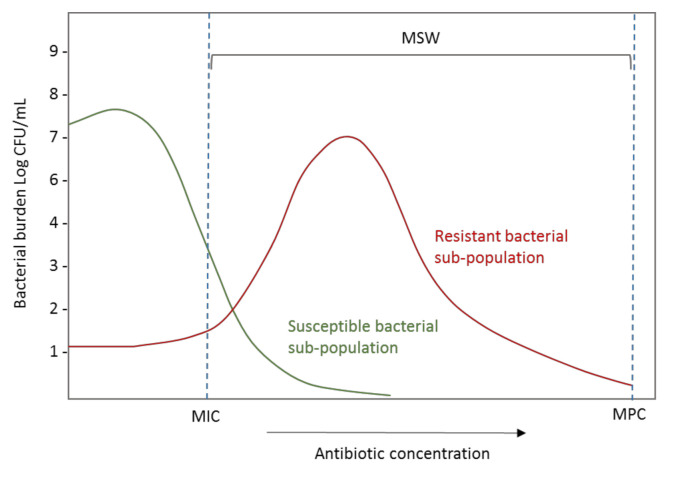
Effect of exposure to the increasing antibiotic concentration on the burden of resistant and susceptible bacterial populations. CFU: Colony-forming units; MIC: Minimum inhibitory concentration; MPC: Mutant prevention concentration; MSW: Mutation selection window.

**Table 1 pharmaceutics-13-00833-t001:** PD indices related to the efficacy of different antimicrobials.

Antimicrobial Activity		PK/PD Index	
Concentration-dependent activity			
Aminoglycosides			*f*Cmax/MIC
Quinolones			*f*AUC_24_/MIC
Time-dependent activity			
β-lactams			*f*T_>MIC_
Penicillins			
Cephalosporins			
Carbapenems			
Concentration-dependent activity with time-dependence			
Vancomycin	Fosfomycin		*f*AUC_24_/MIC
Linezolid	Fluoroquinolones		
Daptomycin	Colistin		

*f*Cmax/MIC: Free-drug maximum concentration to the MIC; *f*T_>MIC_: The percentage of time that the antimicrobial free serum concentration remained above the MIC; *f*AUC_24_/MIC: The area under the free concentration-time curve over 24 h divided by the MIC.

**Table 2 pharmaceutics-13-00833-t002:** Published studies that review breakpoints by applying the PK/PD criteria.

Reference	Bacteria	Antimicrobials	
		Betalactams	Others
DeRyke et al. [64]	*P. aeruginosa* *A. baumanii* *E. coli* *Klebsiella* spp.	Cefepime Ceftazidime Ceftriaxona Imipenem Meropenem Piperacillin/tazabactam	Ciprofloxacin Levofloxacin
Frei et al. [65]	Enterobateriaceae *P. aeruginosa* *A. baumannii*	Aztreonam Cefepime Ceftizoxime Cetazidime Ertapenem Imipenem Meropenem Piperacillin/tazobactam	Ciprofloxacin Gentamicin Levofloxacin Tobramycin
Asín et al. [61]	*Enterococcus* *Staphylococcus* β-Haemolytic streptococci Other streptococci *S. pneumoniae*	Amoxicillin Cefepime Cefotaxime Cloxacillin Ertapenem Imipenem Meropenem Piperacillin/tazobactam	Levofloxacin Vancomycin Daptomycin Tigecycline Linezolid
Burgess et al. [66]	*Neisseria meningitidis*	Ampicillin Cefotaxime Ceftriaxone Ciprofloxacin Meropenem Penicillin G	Azithromycin Chloramphenicol Doxycycline Levofloxacin Minocycline Rifampicin Sulphafurazole Tetracycline Co-Trimoxazole
Zuur et al. [68]	*Mycobacterium tuberculosis*		Isoniazid Pyrazinamide Rifampin
Deshpande et al. [67]	*Mycobacterium tuberculosis*		Levofloxacin

**Table 3 pharmaceutics-13-00833-t003:** PBPK models developed with antimicrobial drugs in special populations.

Reference	Patient Population	Antimicrobial	Route of Administration
Balbas-Martinez et al. [103]	Children with complicated urinary tract infection	Ciprofloxacin	Oral/intravenous
Schlender et al. [104]	Pediatric/adult/geriatric	Ciprofloxacin	Oral/intravenous
Montanha et al. [105]	Bariatric patients	Amoxicillin	Oral tablet/suspension
Thémans et al. [106]	Critical/non critical/obese	Meropenem	Intravenous
Cordes et al. [107]	Patients with tuberculosis	Isoniazid	Oral
Hornik et al. [108]	Pediatric	Clindamycin	Intravenous
Rimmler et al. [109]	Perioperative patients	Cefuroxime	Intravenous
Joyner et al. [110]	Patients with different body mass indexes	Ertapenem	Intravenous
Tod et al. [111]	Patients with hemorrhagic shock followed by fluid resuscitation	Amoxicillin-clavulanate	Intravenous

**Table 4 pharmaceutics-13-00833-t004:** Published articles about PK/PD modeling in veterinary medicine.

Reference	Year	Antimicrobial	Study Description
Burch et al. [136]	2018	Amoxicillin	Review the PK and PD in pigs
El Badawy et al. [137]	2019	Cefquinome	PK and PD in lactating goats
Lei et al. [138]	2018	Piscidin	Evaluation of an antimicrobial peptide in a rat animal model for a future use in veterinary medicine
Vercelli et al. [139]	2020	Levofloxacin	PK/PD of levofloxacin in non-lactating goats
Birhanu et al. [140]	2020	Marbofloxacin in combination with methyl gallate	PK/PD analysis of marbofloxacin in combination with methyl gallate in a rat animal model
Li et al. [141]	2020	Colistin in Combination With Gamithromycin	In vitro susceptibility and time-kill tests and in vivo PK and PD assays using a neutropenic murine lung infection model
Zeng et al. [142]	2018	Tildipirosin	PK/PD modeling in a murine lung infection model
Maan et al. [143]	2020	Aditoprim	In vivo intrauterine PK in cattles and in vitro and ex vivo PD
Huang et al. [144]	2019	Tilmicosin	PK/PD analysis in an in vitro dynamic model
Fernández-Varón et al. [145]	2016	Ceftiofur	PK in lactating goats, in vitro and ex vivo activity, and determination of PK/PD index
Yang et al. [146]	2021	Danofloxacin	PK in piglet and PK/PD analysis in vivo and ex vivo
Yu et al. [147]	2017	Sarafloxacin	PK/PD in Muscovy ducks
Cazer et al. [148]	2017	Chlortetracycline	PK/PD and the enteric bacterial population dynamics in beef cattle

## Data Availability

Not applicable.

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
