# Peer review of "The Role of PK/PD Analysis in the Development and Evaluation of Antimicrobials"

_pharmaceutics, 2021, doi:10.3390/pharmaceutics13060833_

Round 1

Reviewer 1 Report

173 include the lowest antibacterial agent’s concentration that does not kill any bacteria as
174 well as the highest concentration that completely kills the bacteria.

Obs.  Definitions are incorrect. Following these definitions, the smallest is zero and the largest is infinite; please check.

477 i.e., pharmacometric modeling incorporating biopharmaceutical principles has demon478
started (?) to be a valuable tool in drug discovery and development

Obs. Please give some examples of biopharmaceutical principles

487 and comparing the results with CLSI and EUCAST

Obs. It is not explained what does mean CLSI and EUCAST

ad509
justing the dosing regimen. Exposure is a function of how the mode of administration,
510 dose, dosing interval, infusion time, as well as distribution and excretion of the antimi511
crobial agent will influence the infecting organism at the site of infection [15].

Obs. The sentence is unclear. Please check.

Reviewer 2 Report

In the manuscript entitled “The role of PK/PD analysis in the development and evaluation of antimicrobials”, the authors reported the PK/PD principles and the models to investigate the relationship between the PK and the PD of antibiotics. In addition, the authors also describe the role of the PK/PD analysis at different levels, from the development and evaluation of new antibiotics to the optimization of the dosage regimens of currently available drugs, both for human and animal use. Overall, this is an interesting and well-written review and will have potential impact in the current scenario. I have few suggestions to improve the quality and gain more attraction of this manuscript.

Comments:

  1. The authors described about various in vitro models, but how about their correlation with in vivo studies? Authors may consider including a statement.
  2. It will be interesting to have more description about the expectations from PK-PD modelling and simulation.
  3. Authors may provide more discussion regarding limitations and challenges.
  4. Including more description regarding regulatory experience could be interesting, though not necessary.
  5. Is it possible to explore a relationship between PK and PD using the basic pharmacokinetic data and in vitro susceptibility profiles of micro-organisms?
